# Neural Compositional Denotational Semantics for Question Answering

## Abstract

Answering compositional questions requiring multi-step reasoning is challenging for current models. We introduce an end-to-end differentiable model for interpreting questions, which is inspired by formal approaches to semantics. Each span of text is represented by a denotation in a knowledge graph, together with a vector that captures ungrounded aspects of meaning. Learned composition modules recursively combine constituents, culminating in a grounding for the complete sentence which is an answer to the question. For example, to interpret *not green*, the model will represent *green* as a set of entities, *not* as a trainable ungrounded vector, and then use this vector to parametrize a composition function to perform a complement operation. For each sentence, we build a parse chart subsuming all possible parses, allowing the model to jointly learn both the composition operators and output structure by gradient descent. We show the model can learn to represent a variety of challenging semantic operators, such as quantifiers, negation, disjunctions and composed relations on a synthetic question answering task. The model also generalizes well to longer sentences than seen in its training data, in contrast to LSTM and RelNet baselines. We will release our code.

## 1 Introduction

Compositionality is a mechanism by which the meanings of complex expressions are systematically determined from the meanings of their parts, and has been widely assumed in the study of both natural languages (Montague, 1973), as well as programming and logical languages, as a means for allowing speakers to generalize to understanding an infinite number of sentences. Popular neural network approaches to question answering use a restricted form of compositionality, typically encoding a sentence word-by-word from left-to-right, and finally executing the complete sentence encoding against a knowledge source (Perez et al., 2017). Such models can fail to generalize from training sentences in surprising ways. Inspired by linguistic theories of compositional semantics, we instead build a latent tree of interpretable expressions over a sentence, recursively combining constituents using a small set of neural modules. When tested on longer questions than are found in the training data, we find that our model achieves higher performance than baselines using LSTMs and RelNets.

Our approach resembles Montague semantics, in which a tree of interpretable expressions is built over the sentence, with nodes combined by a small set of composition functions. However, both the structure of the sentence and the neural modules that handle composition are learned by end-to-end gradient descent. To achieve this, we define the parametric form of small set of neural modules, and then build a parse chart over each sentence subsuming all possible trees. Each node in the chart represents a span of text with a distribution over groundings (in terms of booleans and knowledge base nodes and edges), as well as a vector representing aspects of the meaning that have not yet been grounded. The representation for a node is built by taking a weighted sum over different ways of building the node (similarly to Maillard et al. (2017)).

Typical neural network approaches to grounded question answering first encode a question from left-to-right with a recurrent neural network (RNNs), and then evaluate the encoding against an encoding of the knowledge source (for example, a knowledge base or image) (Santoro et al., 2017). In contrast to classical approaches to compositionality, constituents of complex expressions are not given explicit interpretations in isolation. For example, in *Which cubes are large or green?*, an RNN encoder will not explicitly build an interpretation for the expression *large or green*. We show that

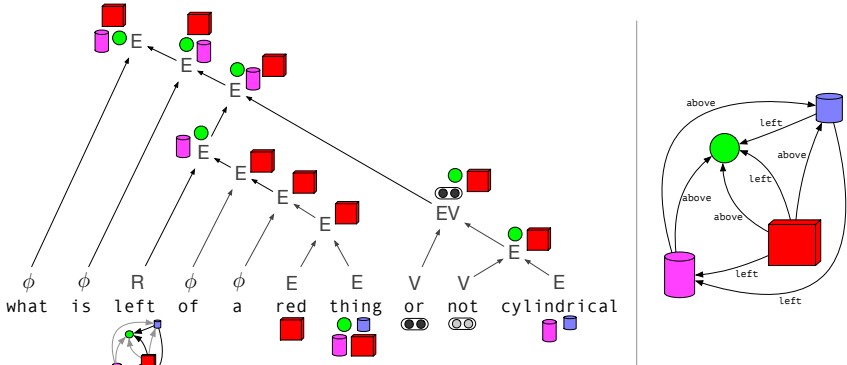

Figure 1: A correct parse for a question given the knowledge graph on the right, using our model. We show the type for each node, and its denotation in terms of the knowledge graph. The words *or* and *not* are represented by vectors, which parameterize composition modules. The denotation for the complete question represents the answer to the question. Nodes here have types $E$ for sets of entities, $R$ for relations, $V$ for ungrounded vectors, $EV$ for a combination of entities and a vector, and $\phi$ for semantically vacuous nodes. While we show only one parse tree here, our model builds a parse chart subsuming all trees.

such approaches can generalize poorly when tested on more complex sentences than they were trained on. In contrast, our approach imposes strong independence assumptions that give a linguistically motivated inductive bias. In particular, it enforces that phrases are interpreted independently of surrounding words, allowing the model to generalize naturally to interpreting phrases in different contexts. In the previous example, *large or green* will be represented as a particular set of entities in a knowledge graph, and be intersected with the set of entities represented by the *cubes* node.

Another perspective on our work is as a method for learning the layouts of Neural Module Networks (NMNs) (Andreas et al., 2016b). Work on NMNs has focused on how to construct the structure of the network, variously using rules, parsers and reinforcement learning (Andreas et al., 2016a; Hu et al., 2017). Our end-to-end differentiable model jointly learns structures and modules by gradient descent.

## 2 MODEL OVERVIEW

Our task is to answer a question $q = w_{1..|q|}$, with respect to a Knowledge Graph (KG) consisting of nodes $\mathcal{E}$ (representing entities) and labelled directed edges $\mathcal{R}$ (representing relationship between entities). In our task, answers are either booleans, or specific subsets of nodes from the KG.

Our model builds a parse for the sentence, in which phrases are grounded in the KG, and a small set of composition modules are used to combine phrases, resulting in a grounding for the complete question sentence that answers the question. For example, in Figure 1, the phrases *not* and *cylindrical* are interpreted as a function word and an entity set, and then *not cylindrical* is interpreted by computing the complement of the entity set. The node at the root of the parse tree is the answer to the question.

We describe a compositional neural model that answers such questions by:

1. Grounding individual tokens in a Knowledge Graph. Tokens can either be grounded as particular sets of entities and relations in the KG, as ungrounded vectors, or marked as being semantically vacuous. For each word, we learn parameters that are used to compute a distribution over semantic types and corresponding denotations in a KG (§ 4.1).

2. Combining representations for adjacent phrases into representations for larger phrases, using trainable neural composition modules (§ 3.2). This produces a denotation for the phrase.

3. Assigning a binary-tree structure to the question sentence, which determines how words are grounded, and which phrases are combined using which modules. We build a parse chart subsuming all possible structures, and train a parsing model to increase the likelihood of structures leading to the correct answer to questions. Different parses leading to a denotation for a phrase of type $t$ are merged into an expected denotation, allowing dynamic programming (§ 4).

4. Answering the question, with the most likely grounding of the phrase spanning the sentence.

## 3  COMPOSITIONAL SEMANTICS

### 3.1  SEMANTIC TYPES

Our model classifies spans of text into different semantic types to represent their meaning as explicit denotations or ungrounded vectors. All phrases are assigned a distribution over semantic types. The semantic type determines how a phrase is grounded, and which composition modules can be used to combine it with other phrases. A phrase spanning $w_{i..j}$ has a denotation $[\![w_{i..j}]\!]_{KG}^{t}$ for each semantic type $t$. For example, in Figure 1, *red thing* corresponds to a set of entities, *left* corresponds to a set of relations, and *not* is treated as an ungrounded vector.

The semantic types we define can be classified into the three different categories. Below we describe these semantic types and their corresponding representations.

**Grounded Semantic Types:**  Spans of text that can be fully grounded in the KG.

1. **Entity** (**E**): Spans of text that can be grounded to a set of entities in the KG, for example: *red sphere* or *large cube*. **E**-type span grounding is represented as a soft-attention value for each entity, $[p_{e_1}, \ldots, p_{e_{|\mathcal{E}|}}]$, where $0 \leq p_{e_i} \leq 1$. This can be viewed as a soft version of a logical set-valued denotation, which we refer to as a 'soft entity set'.

2. **Relation** (**R**): Spans of text that can be grounded to a set of relations from the KG, for example: *left of* or *not right of or above*. **R**-type span grounding is represented by a soft adjacency matrix $A \in \mathbb{R}^{|\mathcal{E}| \times |\mathcal{E}|}$ where $A_{ij} = 1$ denotes a directed edge from $e_i \to e_j$.

3. **Truth** (**T**): Spans of text that can be interpreted as having a True/False denotation, for example: *Is anything red?*, *Is one ball green and are no cubes red?* **T**-type span grounding is represented using a real-value $p_{true}$, $0 \leq p_{true} \leq 1$, that denotes the probability of the span being True.

**Ungrounded Semantic Types:**  Spans of text whose meaning cannot be grounded in the KG.

1. **Vector** (**V**): This type is used for spans representing functions that cannot yet be grounded in the KG, for example words such as *and* or *every*. These spans are represented using 4 different real-valued vectors $v_1 \in \mathbb{R}^2, v_2 \in \mathbb{R}^3, v_3 \in \mathbb{R}^4, v_4 \in \mathbb{R}^5$ that are used to parameterize different composition modules described below in § 3.2.

2. **Vacuous** ($\phi$): Spans that are considered semantically vacuous, but are necessary syntactically, e.g. *of* in *left of a cube*. During composition, these nodes act as identity functions.

**Partially-Grounded Semantic Types:**  Spans of text that can only be partially grounded in the knowledge graph, such as *and red* or *are four spheres*. Here, we represent the span by a combination of a grounding and vectors, representing grounded and ungrounded aspects of meaning respectively. The grounded component of the representation will typically combine with another fully grounded representation, and the ungrounded vectors will parameterize the composition module. We define 3 semantic types of this kind: **EV**, **RV** and **TV**, corresponding to the combination of entities, relations and boolean groundings with an ungrounded vector. Here, the word represented by the vectors can be viewed as a binary function, one of whose arguments has been supplied.

### 3.2  COMPOSITION MODULES

Next, we describe how we compose phrase representations (from § 3.1) to create representations for larger phrases. We define a small number of generic composition modules, that take as input two constituents of text with their corresponding semantic representations (grounded representations and ungrounded vectors), and outputs the semantic type and corresponding representation of the larger constituent. The composition modules are parameterized by the trainable word vectors.

These can be divided into several categories:

**Composition modules resulting in fully grounded denotations:**  Described in Figure 2.

$$p_{e_i} = \sigma\left(\begin{bmatrix} w_1 \\ w_2 \\ b \end{bmatrix} \cdot \begin{bmatrix} p_{e_i}^L \\ p_{e_i}^R \\ 1 \end{bmatrix}\right)$$

**E + E → E**: This module performs a function on a pair of soft entity sets, parameterized by the model's global parameter vector $[w_1, w_2, b]$ to produce a new soft entity set. The composition function for a single entity's resulting attention value is shown. Such a composition module can be used to interpret compound nouns and entity appositions. For example, the composition module shown above learns to output the intersection of two entity sets.

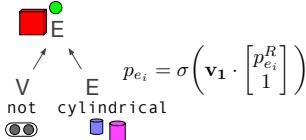

$$p_{e_i} = \sigma\left(\mathbf{v_1} \cdot \begin{bmatrix} p_{e_i}^R \\ 1 \end{bmatrix}\right)$$

**V + E → E**: This module performs a function on a soft entity set, parameterized by a word vector, to produce a new soft entity set. For example, the word *not* learns to take the complement of a set of entities. The entity attention representation of the resulting span is computed by using the indicated function that takes the $v_1 \in \mathbb{R}^2$ vector of the **V** constituent as a parameter argument and the entity attention vector of the **E** constituent as a function argument.

$$p_{e_i} = \sigma\left(\mathbf{v_2} \cdot \begin{bmatrix} p_{e_i}^L \\ p_{e_i}^R \\ 1 \end{bmatrix}\right)$$

**EV + E → E**: This module combines two soft entity sets into a third set, parameterized by the $v_2$ word vector. This composition function is similar to a linear threshold unit and is capable of modeling various mathematical operations such as logical conjunctions, disjunctions, differences etc. for different values of $v_2$. For example, the word *or* learns to model set union.

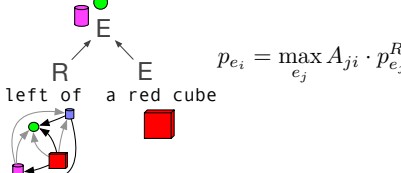

$$p_{e_i} = \max_{e_j} A_{ji} \cdot p_{e_j}^R$$

**R + E → E**: This module composes a set of relations (represented as a single soft adjacency matrix) and a soft entity set to produce an output soft entity set. The composition function uses the adjacency matrix representation of the **R**-span and the soft entity set representation of the **E**-span.

$$p_{true} = \sigma\left(v_3^1 \left[\sum_{e_i} \sigma\left(\begin{bmatrix} v_3^3 \\ v_3^4 \\ v_3^5 \end{bmatrix} \cdot \begin{bmatrix} p_{e_i}^R \\ 1 \end{bmatrix}\right)\right] + v_3^2\right)$$

**V + E → T**: This module maps a soft entity set onto a soft boolean, parameterized by word vector ($v_3$). The module counts whether a sufficient number of elements are in (or out) of the set. For example, the word *any* should test if a set is non-empty.

$$p_{true} = \sigma\left(v_4^1 \left[\sum_{e_i} \sigma\left(\begin{bmatrix} v_4^3 \\ v_4^4 \\ v_4^5 \end{bmatrix} \cdot \begin{bmatrix} p_{e_i}^L \\ p_{e_i}^R \\ 1 \end{bmatrix}\right)\right] + v_4^5\right)$$

**EV + E → T**: This module combines two soft entity sets into a soft boolean, which is useful for modelling generalized quantifiers. For example, in *is every cylinder blue*, the module can use the inner sigmoid to test if an element $e_i$ is in the set of cylinders ($p_{e_i}^L \approx 1$) but not in the set of blue things ($p_{e_i}^R \approx 0$), and then use the outer sigmoid to return a value close to 1 if the sum of elements matching this property is close to 0.

$$p_{true} = \sigma\left(\mathbf{v_2} \cdot \begin{bmatrix} p_{true}^L \\ p_{true}^R \\ 1 \end{bmatrix}\right)$$

**TV + T → T**: This module maps a pair of soft booleans into a soft boolean using the $v_2$ word vector to parameterize the composition function. Similar to **EV + E → E**, this module facilitates modeling a range of boolean set operations. Using the same functional form for different composition functions, allows our model to use the same ungrounded word vector ($v_2$) for compositions that are semantically analogous.

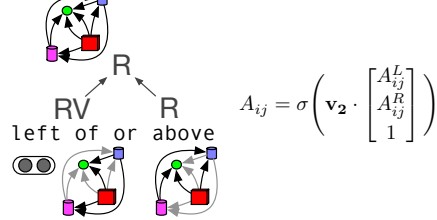

$$A_{ij} = \sigma\left(\mathbf{v_2} \cdot \begin{bmatrix} A_{ij}^L \\ A_{ij}^R \\ 1 \end{bmatrix}\right)$$

**RV + R → R**: This module composes a pair of soft set of relations to a produce an output soft set of relations. For example, the relations *left* and *above* are composed by the word *or* to produce a set of relations such that entities $e_i$ and $e_j$ are related if either of the two relations exists between them. The functional form for this composition is similar to **EV + E → E** and **TV + T → T** modules.

Figure 2: Composition Modules that compose two constituent span representations into the representation for the combined larger span, using the indicated equations.

**Composition with $\phi$-typed nodes:** Phrases with type $\phi$ are treated as being semantically transparent identity functions. Phrases of any other type can combined with these with no change to their type or representation.

**Composition modules resulting in partially grounded denotations:** We define several simple modules that combine fully grounded phrases with ungrounded phrases, by deterministically taking the union of the representations, giving phrases with partially grounded representations (§ 3.1). These modules are useful for when words act as binary functions; here they combine with their first argument. For example, in Figure 1, *or* and *not cylindrical* combine to make a phrase containing both the vectors for *or* and the entity set for *not cylindrical*.

# 4 PARSING MODEL

Here, we describe how our model classifies question tokens into different semantic type spans and compute their representations (§ 4.1), recursively uses the composition modules defined above to parse the question appropriately into a soft latent tree that provides the answer (§ 4.2). The model is trained end-to-end using only question-answer supervision (§ 4.3).

## 4.1 LEXICAL REPRESENTATION ASSIGNMENT

Each token in the question sentence is assigned a distribution over the semantic types, and given a grounding for each type. Tokens can only be assigned the **E**, **R**, **V**, and $\phi$ semantic types. For example, the token *cylindrical* in the question in Fig. 1 is assigned a distribution over the 4 semantic types (one shown) and for the **E** type, the representation computed is the set of *cylindrical* entities.

**Semantic Type Distribution for Tokens:** To compute the semantic type distribution, our model represents each word $w$ in the word vocabulary $\mathcal{V}$, and each semantic type $t$ using an embedding vector; $v_w, v_t \in \mathbb{R}^d$. The semantic type distribution is assigned with a softmax:

$$p(t|w_i) \propto \exp(v_t \cdot v_{w_i}) \tag{1}$$

**Grounding for Tokens:** For each of the four semantic type assignments for question tokens, we need to compute/assign their corresponding representations.

1. **E-Type Representation:** Each entity $e \in \mathcal{E}$, is represented using an embedding vector $v_e \in \mathbb{R}^d$ based on the concatenation of vectors for its properties. For each token $w$, we use its word vector to find the probability of each entity being part of the **E**-Type grounding:

$$p_{e_i}^w = \sigma(v_{e_i} \cdot v_w) \ \ \forall \, e_i \in \mathcal{E} \tag{2}$$

   For example, in Fig. 1, the word *red* will be grounded as all the red entities.

2. **R-Type Representation:** Each relation $r \in \mathcal{R}$, is represented using an embedding vector $v_r \in \mathbb{R}^d$. For each token $w_i$ in the question, we first compute a distribution over relations it could refer to, and then use this distribution to compute the *expected* adjacency matrix that forms the **R**-type representation for this token.

$$p(r|w_i) \propto \exp(v_r \cdot v_{w_i}) \tag{3}$$

$$A^{w_i} = \sum_{r \in \mathcal{R}} p(r|w_i) \cdot A_r \tag{4}$$

   For example, the word *left* in Fig. 1 is grounded as the subset of edges with the label 'left'.

3. **V-Type Representation:** For each word $w \in \mathcal{V}$, we learn four vectors $v_1 \in \mathbb{R}^2, v_2 \in \mathbb{R}^3, v_3 \in \mathbb{R}^4, v_4 \in \mathbb{R}^5$, and use these as the representation for words with the **V**-Type.

4. **$\phi$-Type Representation:** This type is used for semantically vacuous words, which do not require a representation.

## 4.2 PARSING QUESTIONS

To learn the correct structure for applying composition modules, we use a simple parsing model. We build a parse-chart over the question encompassing all possible trees by applying all composition modules, similar to a standard CRF-based PCFG parser using the `CKY` algorithm. Each node in the parse-chart, for each span $w_{i..j}$ of the question, is represented as a distribution over different semantic types with their corresponding representations. This distribution is computed by weighing the different ways of composing the span's constituents.

**Phrase Semantic Type Potential:** Each node in the parse-chart is associated with a potential value $\psi(i, j, t)$, that is the score assigned by the model to the **t** semantic type for the $w_{i..j}$ span. This is computed from all possible ways to form the span $w_{i..j}$ with type **t**. For a particular composition of span $w_{i..k}$ of type $\mathbf{t_1}$ and $w_{k+1..j}$ of type $\mathbf{t_2}$, using the $\mathbf{t_1} + \mathbf{t_2} \rightarrow \mathbf{t}$ module, the score is:

$$\psi(i, j, k, t_1 + t_2 \rightarrow t) = \left[ \psi(i, k, t_1) \cdot \psi(k+1, j, t_2) \cdot \exp\left( \sum_x f_x^{(t_1 + t_2 \rightarrow t)}(i, j, k | q) \right) \right] \quad (5)$$

where, $f_x^{(t_1 + t_2 \rightarrow t)}(i, j, k | q)$ are six feature functions; a trainable weight for each word per module in the vocabulary, that correspond to: $f_1$: word that appears before the start of the span $w_{i-1}$; $f_2$: first word in the span $w_i$; $f_3$: last word in the left constituent $w_k$; $f_4$: first word in the right constituent $w_{k+1}$; $f_5$: last word in the right constituent $w_j$; and $f_6$: word that appears after the span $w_{j+1}$.

The token semantic type potential of $w_i$, $\psi(i, i, i, t_1 + t_2 \rightarrow t)$, is the same as $p(t|w_i)$ (Eq. 1).

The final **t**-type potential of $w_{i..j}$ is computed by summing over scores from all possible compositions:

$$\psi(i, j, t) = \sum_{k=i}^{j-1} \sum_{\substack{(t_1 + t_2 \rightarrow t) \\ \in \text{Modules}}} \psi(i, j, k, t_1 + t_2 \rightarrow t) \quad (6)$$

**Combining Phrase Representations:** To compute the span $w_{i..j}$'s denotation with type **t**, $[\![w_{i..j}]\!]_{KG}^t$, we compute an expected output representation from all possible compositions.

$$[\![w_{i..j}]\!]_{KG}^t = \frac{1}{\psi(i, j, t)} \sum_{k=i}^{j-1} \sum_{\substack{(t_1 + t_2 \rightarrow t) \\ \in \text{Modules}}} \psi(i, j, k, t_1 + t_2 \rightarrow t) * [\![w_{i..k..j}]\!]_{KG}^{t_1 + t_2 \rightarrow t} \quad (7)$$

where $[\![w_{i..j}]\!]_{KG}^t$, is the **t**-type representation of the span $w_{i..j}$, $[\![w_{i..k..j}]\!]_{KG}^{t_1 + t_2 \rightarrow t}$ is the representation resulting from the composition of $w_{i..k}$ with $w_{k+1..j}$ using the $\mathbf{t_1} + \mathbf{t_2} \rightarrow \mathbf{t}$ composition module.

**Answer Grounding:** By recursively computing the phrase semantic-type potentials and representations, we can infer the semantic type distribution of the complete question sentence (Eq. 8) and the resulting grounding for different semantic type $t$, $[\![w_{1..|q|}]\!]_{KG}^t$.

$$p(t|q) \propto \psi(1, |q|, t) \quad (8)$$

The answer-type (boolean or subset of entities) for the question is computed using:

$$t^* = \underset{t \in \mathbf{T}, \mathbf{E}}{\text{argmax}} \ p(t|q) \quad (9)$$

The corresponding grounding is $[\![w_{1..|q|}]\!]_{KG}^{t^*}$, which answers the question.

## 4.3 TRAINING OBJECTIVE

Given a dataset $\mathcal{D}$ of (question, answer, knowledge-graph) tuples, $\{q^i, a^i, KG^i\}_{i=1}^{i=|\mathcal{D}|}$, we train our model to maximize the log-likelihood of the correct answers. Answers are either booleans, or specific subsets of entities from the KG. We denote the semantic type of the answer as $a_t$. If the answer is boolean, $a \in \{0, 1\}$, otherwise is a subset of entities from the KG, i.e. $a = \{e_j\}$. The model's answer to a question is found by taking its representation of the complete question, containing a distribution over types and the representation for each type. We maximize the following objective:

$$\mathcal{L} = \sum_i \log p(\mathbf{a}^i | \mathbf{q}^i, \mathbf{KG}^i) \tag{10}$$

$$= \sum_i \left[ \underbrace{\left( \mathbb{1}_{a_t^i = \mathbf{T}} \left[ \log(p_{true})^{a^i} (1 - p_{true})^{(1-a^i)} \right] \right)}_{\text{Questions with boolean answers}} + \underbrace{\left( \frac{\mathbb{1}_{a_t^i = \mathbf{E}}}{|\mathcal{E}^i|} \left[ \log \prod_{e_j^i \in a^i} p_{e_j^i} \prod_{e_j^i \notin a^i} (1 - p_{e_j^i}) \right] \right)}_{\text{Questions with entity set answers}} \right]$$
$$\tag{11}$$

We also add $L_2$-regularization for the scalar parsing features introduced in § 4.2.

## 5 DATASET

We generate a dataset of question-answers based on the CLEVR dataset (Johnson et al., 2017), which contains knowledge graphs containing attribute information of objects and relations between them.

We generate a new set of questions for this data, as existing questions contain some biases that can be exploited by models (Johnson et al. (2017) found that many spatial relation questions can be answered only using absolute spatial information and many long questions can be answered correctly without performing all steps of reasoning), and many questions are over 40 words long, which is intractable given that the size of our computation graph is cubic in the question length. Future work should explore scaling our approach to longer questions. We generate 75K questions for training and 37.5K for validation.

Our question set tests various challenging semantic operators. These include conjunctions (e.g. *Is anything red or is anything large?*), negations (e.g. *What is not spherical?*), counts (e.g. *Are five spheres green?*), quantifiers (e.g. *Is every red thing cylindrical?*), and relations (e.g. *What is left of and above a cube?*). We employ some simple tests to remove trivial biases from the dataset.

We create two test sets: one drawn from the same distribution as the training data (37.5K), and another containing longer questions than the training data (22.5K).

Our COMPLEX QUESTIONS test set contains the same words and constructions, but chained into longer questions. For example, it contains questions such as *What is a cube that is right of a metallic thing that is beneath a blue thing?* and *Are two red things that are above a sphere metallic?*. These questions require more multi-step reasoning to solve.

## 6 EXPERIMENTS

In this section we describe our experimentation setting, the baseline models we compare to, and the various experiments demonstrating the ability of our model to answer compositional questions referring to KG and its ability to generalize to unseen longer questions and new attribute combinations.

### 6.1 EXPERIMENTATION SETTING

Here we describe the training details of our model and the baseline models.

**Representing Entities:** Each entity in the CLEVR dataset consists of 4 attributes. For each attribute-value, we learn an embedding vector and concatenate the 4-embedding vectors to form the representation for the entity.

**Training Details:** Training the model is complicated by the large number of poor local minima, as the model needs to learn both good syntactic structures and the complex semantics of neural modules.

To simplify training, we use Curriculum Learning (Bengio et al., 2009) to pre-train the model on an easier subset of questions. We use a 2-step schedule where we first train our model on simple attribute match (*What is a red sphere?*), attribute existence (*Is anything blue?*) and boolean composition (*Is anything green and is anything purple?*) questions and in the second step on all questions jointly.

| Model | Boolean Questions | Entity Set Questions | Relation Questions | Overall |
|---|---|---|---|---|
| LSTM (NO KG) | 50.7 | 14.4 | 17.5 | 27.2 |
| LSTM (NO RELATION) | 88.5 | 99.9 | 15.7 | 84.9 |
| RELATION NETWORK | 85.6 | 89.7 | 97.6 | 89.4 |
| Our Model | 99.9 | 100 | 100.0 | 99.9 |

Table 1: **Results for Short Questions**: Performance of our model compared to baseline models on the Short Questions test set. The LSTM (NO KG) has accuracy close to chance, showing that the questions lack trivial biases.Our model almost perfectly solves all questions showing its ability to learn challenging semantic operators, and parse questions only using weak end-to-end supervision.

We tune the hyper-parameters using validation accuracy. We train using SGD with learning rate of $0.5$ and mini-batch size of 4, regularization constant of $0.3$. When assigning the semantic type distribution to the words at the leaves, we add a small positive bias of $+1$ for $\phi$-type and a small negative bias of $-1$ for the **E**-type score before the softmax. Our trainable parameters are: question word embeddings (64-dimensional), relation embeddings (64-dimensional), entity attribute-value embeddings (16-dimensional), four vectors per word for **V**-type representations, six scalar feature scores per module per word for the parsing model, and the global parameter vector for the **E+E→E** module.

**Baseline Models:**   We use three baseline models for comparison. A simple LSTM (NO KG) model that encodes the question using an LSTM network and answers questions without access to the KG. Another LSTM based model, LSTM (NO RELATION), that has access only to the entities of the KG but not the relationship information between them. Finally, we train a RELATION NETWORK (Santoro et al., 2017) augmented model, which achieved state-of-the-art performance on the CLEVR dataset using image state descriptions. Details about the baseline models are given in the Appendix section.

## 6.2   EXPERIMENTS

**Short Questions Performance:**   In Table 1, we see that our model is able to perfectly answer all the questions in the test set. This demonstrates our model can learn challenging semantic operators using composition modules, as well as learn to parse the questions from only using weak end-to-end supervision. The RELATION NETWORK also achieves good performance, particularly on questions involving relations, but is weaker than our model on some question types. The LSTM (NO RELATION) model also achieves good performance on questions not involving relations, which are out of scope for the model.

**Complex Question Performance:**   Table 2 shows results on complex questions, which are constructed by combining components of shorter questions. We use the same models as in Table 1, which were trained and developed only on shorter questions. Answering longer questions requires complex multi-hop reasoning, and the ability to generalize from the language seen in its training data to new types of questions. Results show that all baselines achieve close to random performance on this task, despite high accuracy for shorter questions. This shows the challenges in generalizing RNN encoders beyond their training data. In contrast, the strong inductive bias from our model structure allows the model to generalize to complex questions much more easily than RNN encoders.

**Generalization to Unseen Attribute Combination:**   We also measure how well models generalize to unseen attribute combinations in knowledge graphs (using the COGENT subset of CLEVR). For example, the test set contains 'blue spheres' that are not found in the training set. None of the models showed a significant reduction in performance in this setting.

**Error Analysis:**   Analyzing the errors of our model, we find that most errors are due to incorrect assignments of structure, rather than semantic errors from the modules. For example, in the question *Are four red spheres beneath a metallic thing small?*, our model produces a parse where it composes *metallic thing small* into a single node instead of composing *red spheres beneath a metallic thing* into a single node. Future work should use more sophisticated parsing models.

| Model | Non-Relation Questions | Relation Questions | Overall |
|---|---|---|---|
| LSTM (NO KG) | 46.0 | 39.6 | 41.4 |
| LSTM (NO RELATION) | 62.2 | 49.2 | 52.2 |
| RELATION NETWORK | 51.1 | 38.9 | 41.5 |
| Our Model | 81.8 | 85.4 | 84.6 |

Table 2: **Results for Complex Questions**: All baseline models fail to generalize to questions requiring longer chains of reasoning than seen during training. Our model substantially outperforms the baselines, showing its ability to perform complex multi-hop reasoning, and generalize from its training data. Analysis suggests that most errors from our model are due to assigning incorrect structures, not mistakes by the composition modules.

## 7 RELATED WORK

Many approaches have been proposed to perform question-answering against structured knowledge sources. *Semantic parsing* models have attempted to learn structures over pre-defined discrete operators, to produce logical forms that can be executed to answer the question. Early work trained using gold-standard logical forms (Zettlemoyer & Collins, 2005; Kwiatkowski et al., 2010), whereas later efforts have only used answers to questions (Liang et al., 2011; Krishnamurthy & Kollar, 2013; Pasupat & Liang, 2015). A key difference is that our model must learn semantic operators from data, which may be necessary to model the fuzzy interpretations of some function words like *many* or *few*.

Another similar line of work is neural program induction models, such as Neural Programmer (Neelakantan et al., 2016) and Neural Symbolic Machine (Liang et al., 2017). These models learn to produce programs composed of predefined operators using weak supervision to answer questions against semi-structured tables.

Neural module networks have recently been proposed for learning semantic operators (Andreas et al., 2016b) for question answering. This model assumes that the structure of the semantic parse is given, and must only learn a set of operators. Dynamic Neural Module Networks (D-NMN) extend this approach by selecting from a small set of candidate module structures (Andreas et al., 2016a). In contrast, our approach learns a model over all possible structures for interpreting a question.

Our work is most similar to the most recently proposed N2NMN (Hu et al., 2017) model, an end-to-end version of D-NMN. This model learns both semantic operators and the layout in which to compose them. However, optimizing the layouts requires reinforcement learning, which is challenging due to the high variance of policy gradients, whereas our approach is end-to-end differentiable.

## 8 CONCLUSION

We have introduced a model for answering questions requiring compositional reasoning that combines ideas from compositional semantics with end-to-end learning of composition operators and structure. We demonstrated that the model is able to learn a number of complex composition operators from end task supervision, and have shown that the linguistically motivated inductive bias imposed by the structure of the model allows it to generalize well beyond its training data. Future work should explore scaling the model to other question answering tasks.

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

## APPENDIX

### BASELINE MODELS

#### LSTM (NO KG)

We use a LSTM network to encode the question as a vector $q$. We also define three other parameter vectors, $t$, $e$ and $b$ that are used to predict the answer-type $P(a = \mathbf{T}) = \sigma(q \cdot t)$, entity attention value $p_{e_i} = \sigma(q \cdot e)$, and the probability of the answer being True $p_{true} = \sigma(q \cdot b)$.

#### LSTM (NO RELATION)

Similar to LSTM (NO RELATION), the question is encoded using a LSTM network as vector $q$. Similar to our model, we learn entity attribute-value embeddings and represent each entity as the concatenation of the $4$ attribute-value embeddings, $v_{e_i}$. Similar to LSTM (NO RELATION), we also define the $t$ parameter vector to predict the answer-type. The entity-attention values are predicted as $p_{e_i} = \sigma(v_{e_i} \cdot q)$. To predict the probability of the boolean-type answer being true, we first add the entity representations to form $b = \sum_{e_i} v_{e_i}$, then make the prediction as $p_{true} = \sigma(q \cdot b)$.

### RELATION NETWORK AUGMENTED MODEL

The original formulation of the relation network module is as follows:

$$RN(q, KG) = f_\phi \left( \sum_{i,j} g_\theta(e_i, e_j, q) \right) \tag{12}$$

where $e_i$, $e_j$ are the representations of the entities and $q$ is the question representation from an LSTM network. The output of the Relation Network module is a scalar score value for the elements in the answer vocabulary. Since our dataset contains entity-set valued answers, we modified the module in the following manner.

We concatenate the object pair representations with the representations of the pair of directed relationships between them[1]. We then use the Relation Network module to produce an output representation for each entity in the KB, in the following manner:

$$RN_{e_i} = f_\phi \left( \sum_{j} g_\theta(e_i, e_j, r_{ij}^1, r_{ij}^2, q) \right) \tag{13}$$

Similar to the LSTM baselines, we define a parameter vector $t$ to predict the answer-type as:

$$P(a = \mathbf{T}) = \sigma(q \cdot t) \tag{14}$$

$$P(a = \mathbf{E}) = 1 - P(a = \mathbf{T}) \tag{15}$$

To predict the probability of the boolean type answer being true, we define a parameter vector $b$ and predict as following:

$$p_{true} = \sigma \left( b \cdot \sum_{e_i} RN_{e_i} \right) \tag{16}$$

To predict the entity-attention values, we use a separate attribute-embedding matrix to first generate the output representation for each entity, $e_i^{out}$, then predict the output attention values as follows:

$$p_{e_i} = \sigma \left( RN_{e_i} \cdot e_i^{out} \right) \tag{17}$$

We tried other architectures as well, but this modification provided the best performance on the validation set. We also tuned the hyper-parameters and found the setting from Santoro et al. (2017) to work the best based on validation accuracy. We used a different 2-step curriculum to train the RELATION NETWORK module, in which we replace the Boolean questions with the relation questions in the first-schedule and jointly train on all questions in the subsequent schedule.

---

[1]In the CLEVR dataset, between any pair of entities, only 2 directed relations, *left* or *right*, and *above* or *beneath* are present.

