# OpenReview forum: "Neural Compositional Denotational Semantics for Question Answering"
_ICLR.cc/2018/Conference — Reject_

### Official Review · AnonReviewer1 · 2017-11-23

**Rating:** 5
**Confidence:** 4

**Review:**

This paper presents a model for visual question answering that can learn both
parameters and structure predictors for a modular neural network, without
supervised structures or assistance from a syntactic parser. Previous approaches
for question answering with module networks can (at best) make a hard choice
among a small number of structures. By contrast, this approach computes a
bottom-up approximation to the softmax over all possible tree-shaped network
layouts using a CKY-style dynamic program. On a slightly modified set of
structured scene representations from the CLEVR dataset, this approach
outperforms two LSTM baselines with incomplete information, as well as an
implementation of Relation Networks.

I think the core technical idea here is really exciting! But the experimental
validation of the approach is a bit thin, and I'm not ready to accept the paper
in its current form.

RELATED WORK / POSITIONING

The title & first couple of paragraphs in the intro suggest that the
denotational interpretation of the representations computed by the modules is
one of the main contributions of this work. It's worht pointing out that the
connection between formal semantics and these kinds of locally-normalized
"attentions" to entities was already made in the cited NMN papers. Meanwhile,
recent work by Johnson et al. and Perez et al. has found that explicitly
attentional / denotational models are not necessarily helpful for the CLEVR
dataset.

If the current paper really wants to make denotational semantics part of the
core claim, I think it would help to talk about the representational
implications in more detail---what kinds of things can and can't you model
once you've committed to set-like bottlenecks between modules? Are there things
we expect this approach to do better than something more free-form (a la
Johnson)? Can you provide experimental evidence of this?

At the same time, one of these things that's really nice about the
structure-selection part of this model is that it doesn't care what kind of
messages the modules send to each other! It might be just as effective to focus
on the dynamic programming aspect and not worry so much about the semantics of
individual modules.

MODELING

Figure 2 is great. It would be nice to have a little bit of discussion about the
motivation for these particular modeling implementations---some are basically
the same as in Hu et al. (2017), but obviously the type system here is richer
and it might be helpful to highlight some of the extra things it can do.

The phrase type semantic potential seems underpowered relative to the rest of
the model---is it really making decisions on the basis of 6 sparse features for
every (span, type) pair, with no score for the identity of the rule (t_1, t_2 ->
t)? What happens if you use biRNN representations of each anchored token, rather
than the bare token alone? (This is standard in syntactic parsing these days.)
If you tried richer things and found that they didn't help, you should show
ablation experiments.

EXPERIMENTS

As mentioned above, I think this is the only really disappointing piece of this
paper. As far as I know, nobody else has actually worked with the structured KBs
in CLEVR---the whole point of the dataset (and VQA, and the various other recent
question answering datasets) is to get away from requiring structured knowledge
bases. The present experiments involve both fake language data and fake,
structured world representations, so it's not clear how much we should trust the
proposed approach to generalize to real tasks.

We know that more traditional semantic parsing approaches with real logical
forms are capable of getting excellent accuracy on structured QA tasks with a
lot more complexity and less data than this one. I think fairness really
requires a comparison to an approach for semantic parsing with denotations.

But more importantly, why not just run on images? Results on VQA, CLEVR, and
NLVR (even if they're not all state of the art!) would make this paper much more
convincing.

---

> ### Author Response · Authors · 2018-01-05
> **Response to AnonReviewer1**
>
> We thank the reviewer for their helpful review.
> We address the issue regarding the evaluation dataset in a separate official comment.
>
> To clarify, we do not see this work as a VQA model, but as a model for answering questions on knowledge graphs (hence why we don’t run on images). KG question answering is an important task in its own right, so we don’t see the use of a KG as a fake approximation of images. The choice of the CLEVR-style dataset may have been confusing here.
>
> As you say, traditional semantic parsing approaches with hardcoded logical operators would likely work well on this data. However, we are interested in the extent to which these operators can be learnt from scratch with minimal prior knowledge. Also, there are very significant challenges in properly formalizing language in terms of logic, and our work offers a direction for circumventing these issues while still retaining many of the attractive properties of compositional semantics.
>
> In terms of representational power, the use of additional ‘ungrounded’ vectors helps avoid the limitations of set-like bottlenecks, by giving the model another mechanism for passing information. A major advantage compared to fully-freeform sentence representations, we showed that our model offers better generalization to longer questions by having compositionality built in (and also gives a more interpretable output). Compared to Johnson et al., we can learn end-to-end without pre-annotated programs.
>
> Regarding the phrase semantic type potential, we identified a typographical error in the paper and have corrected the same (Eq. 5). The feature function does indeed take into account the identity of the module. Parsing on this data is relatively straightforward, so we did not see additional gains from using RNN models.

---

> > ### Comment · AnonReviewer1 · 2018-01-11
> > **Rebuttal response**
> >
> > Thanks for rebuttal! My concern is that:
> >
> > 1. The underlying representations in the KB version of the dataset are already so clean that the model can't be claimed to be "learning from scratch" in any meaningful sense. At the very least, the problem of lining up a word with the particular one-hot vector that picks out a feature is no more interesting on the surface than the problem of lining up the word with a discrete semantic token.
> >
> > 2. I absolutely agree that there are "significant challenges in properly formalizing language in terms of logic"; the problem is that these problems don't actually show up in this dataset!
> >
> > So minimally, if you're going to use this dataset I think you really have to compare to a regular semantic parser (it would be fine to run UBL or Cornell-SPF out of the box). But it would be even better to use a dataset with real natural language even if you're going to stick to structured world representations.
> >
> > I'm leaving my score as-is for now, but I think this paper is close to ready.

---

### Official Review · AnonReviewer3 · 2017-11-26
**Beautiful and elegant model but evaluation is not satisfying.**

**Rating:** 7
**Confidence:** 4

**Review:**

This paper proposes for training a question answering model from answers only and a KB by learning latent trees that capture the syntax and learn the semantic of words, including referential terms like "red" and also compositional operators like "not".

I think this model is elegant, beautiful and timely. The authors do a good job of explaining it clearly. I like the modules of composition that seem to make a very intuitive sense for the "algebra" that is required and the parsing algorithm is clean.

However, I think that the evaluation is lacking, and in some sense the model exposes the weakness of the dataset that it uses for evaluation.

I have 2.5 major issues with the paper and a few minor comments:

Parsing:

* The authors don't really say what is the base case for \Psi that scores tokens (unless I missed it and if indeed it is missing it really needs to be added) and only provide the recursive case. From that I understand that the only features that they use are whether a certain word makes sense in a certain position of the rule application in the context of the question. While these features are based on Durrett et al.'s neural syntactic parser it seems like a pretty weak signal to learn from. This makes me wonder, how does the parser learn whether one parse is better than the other? Only based on this signal? It makes me suspicious that the distribution of language is not very ambiguous and that as long as you can construct a tree in some context you can do it in almost any other context. This is probably due to the fact that the CLEVR dataset was generated mostly using templates and is not really natural utterances produced by people. Of course many people have published on CLEVR although of its language limitations, but I was a bit surprised that only these features are enough to solve the problem completely, and this makes me curious as to how hard is it to reverse-engineer the way that the language was generated with a context-free mechanism that is similar to how the data was produced.

* Related to that is that the decision for a score of a certain type t for a span (i,j) is the sum for all possible rule applications, rather than a max, which again means that there is no competition between different parse trees that result with the same type of a single span. Can the authors say something about what the parser learns? Does it learn to extract from the noise clear parse trees? What is the distribution of rules in those sums? is there some rule that is more preferred than others usually? It seems like there is loss of information in the sum and it is unclear what is the effect of that in the paper.

Evaluation:

* Related to that is indeed the fact that they use CLEVR only. There  is now the Cornell NLVR dataset that is more challenging from a language perspective and it would be great to have an evaluation there as well. Also the authors only compare to 3 baselines where 2 don't even see the entire KB, so the only "real" baseline is relation net. The authors indeed state that it is state-of-the-art on clevr.

* It is worth noting that relation net is reported to get 95.5 accuracy while the authors have 89.4. They use a subset so this might be the reason, but I am not sure how they compared to relation net exactly. Did they re-tune parameters once you have the new dataset? This could make a difference in the final accuracy and cause an unfair advantage.

* I would really appreciate more analysis on the trees that one gets. Are sub-trees interpretable? Can one trace the process of composition? This could have been really nice if one could do that. The authors have a figure of a purported tree, but where does this tree come from? From the mode? Form the authors?

Scalability:
* How much of a problem would it be to scale this? Will this work in larger domains? It seems they compute an attention score over every entity and also over a matrix that is squared in the number of entities. So it seems if the number of entities is large that could be very problematic. Once one moves to larger KBs it might become hard to maintain full differentiability which is one of the main selling points of the paper.

Minor comments:
* I think the phrase "attention" is a bit confusing - I thought of a distribution over entities at first.
* The feature function is not super clearly written I think - perhaps clarify in text a bit more what it does.
* I did not get how the denotation that is based on a specific rule applycation t_1 + t_2 --> t works. Is it by looking at the grounding that is the result of that rule application?
* Authors say that the neural enquirer and neural symbolic machines produce flat programs - that is not really true, the programs are just a linearized form of a tree, so there is nothing very flat about it in my opinion.

Overall, I really enjoyed reading the paper, but I was left wondering whether the fact that it works so well mostly attests to the way the data was generated and am still wondering how easy it would be to make this work in for more natural language or when the KB is large.

---

> ### Author Response · Authors · 2018-01-05
> **Response to AnonReviewer3**
>
> We thank the reviewer for their helpful review, which will allow us to improve a number of points in the paper.
>
> Parsing:
> * The base case for \Psi is the semantic type distribution for each word computed in Eq. 1. In the updated version we have made this explicitly clear.
> For simplicity, we use a simple feature-based parsing model - although similar features can achieve very good performance on the Penn Treebank. We found that learning with these features was more stable than an RNN parsing model.
> * The score of a certain type t for a span (i,j) is indeed the sum for all possible rule applications, but there is still competition between different parse trees that result with the same type of a single span. This competition arises since the different parse trees result in different denotations (grounding) for this type for the span. Feedback from the root of the parse tree encourages correct grounding and hence correct parse structure.
>
> Evaluation
> * We retrained the RelNet moden on the new dataset, and carefully tuned the hyper-parameters for the RelNet model on validation data.
> * The trees computed by the model are completely interpretable. For each subtree, we can see exactly how the different compositions score  which allows us to completely trace the composition. The tree shown in Fig.1 is the highest scoring tree (mode) from the learned model.
>
> Scalability
> * As you suggest, there would be challenges in scaling the approach to large knowledge graphs, and would require further work to be efficient. As the number of entities grows to an intractable size, KNN search, beam search, feature hashing and parallelization techniques can be explored to make the model tractable. Such techniques are fairly commonly used in large-scale KG QA.
>
> Minor Comments
> * If accepted, we will make this clearer in the camera-ready version.
> * The denotation of a particular rule application t_1 + t_2 → t is indeed the resulting grounding from the module application.

---

### Official Review · AnonReviewer2 · 2017-11-27
**Not convinced by only synthetic data evaluation**

**Rating:** 4
**Confidence:** 4

**Review:**

The paper describes an end to end differentiable model to answer questions based on a knowledge base. They learn the composition modules which combine representations for parts of the question to generate a representation of the whole question.

My major complaint is the evaluation on a synthetically generated data set. Given the method of generating the data, it was not a surprise that the method which leverages hierarchical structure can do better than other methods which do not leverage that. I will be convinced if evaluation can be done on a real data set.

Minor complaints:

The paper does not compare to NMN, or a standard semantic parser. I understand that all other methods will use a predefined set of predicates, but its still worthwhile to see how much we loose when trying to learn predicates from scratch.

The paper mentions that they enumerate all parses. That is true only if the groundings are not considered part of the parse. They actually enumerate all parses based on types, and then find the right groundings for the best parse. This two step inference is an approximation, which should be mentioned somewhere.

Response to rebuttal:

I agree that current data sets have minimal compositionality,  and that "if existing models cannot handle the synthetic data, they will not handle real data". However, its not clear that your method will be better than the alternatives when you move to real data. Also, some work on CLEVR had some questions collected from humans, maybe you can try to evaluate on that. I am going to keep my rating the same.

---

> ### Author Response · Authors · 2018-01-05
> **Response to AnonReviewer2**
>
> We thank the reviewer for their helpful review.
> We address the issue regarding the evaluation dataset in a separate official comment.
> We would like to clarify that we do not follow a two-step procedure and instead compute groundings for each separate parse. The final answer / grounding at the root is the weighted average of groundings from all possible parses. The feedback from the correct answer at the root encourages the grounding and hence the correct parse.

---

### Author Response · Authors · 2018-01-05
**Response to reviewers regarding the evaluation dataset**

We would like to thank all the reviewers for their thoughtful comments and suggestions. We’re glad that they think that “this model is elegant, beautiful and timely” and that the “core technical idea here is really exciting!”

The major concern raised by all the reviewers is the choice of evaluation dataset. We respectfully suggest that the some of the comments are judging the evaluation with respect to claims that we are not making. In our evaluation, we aim to show that the model can simultaneously learn structure and interpretation to perform many-hop reasoning, and that it shows better compositional generalization than alternatives such as LSTMs and RelNets. While it is certainly true that using human language would cause different challenges (primarily due to greater diversity in the language), existing datasets are dominated by simpler questions that do not require the multistep reasoning we focus on. If existing models cannot handle the reasoning involved in the synthetic data we evaluate on, then there is no reason to think they could deal with the additional complexity of human language.

---

### Decision · Program_Chairs · 2018-01-29
**ICLR 2018 Conference Acceptance Decision**

**Decision:**

Reject

**Comment:**

This paper presents a neural compositional model for visual question answering.  The overall idea may be exciting but the committee agrees with the evaluation of Reviewer 1:  the experimental section is a bit thin and it only evaluates against an artificial dataset for visual QA that does not really need a knowledge base.  It would have been better to evaluate on more traditional question answering settings where the answer can be retrieved from a knowledge base (WebQuestions, Free917, etc.), and then compare with state of the art on those.